# Evaluation of Two Surface Renewal Methods for Calculating the Sensible Heat Flux over a Tea Field Ecosystem in Hilly Terrain

**Huijie Hu** [1], **Yongzong Lu** [1,2,*] , **Yongguang Hu** [1] **and Risheng Ding** [3]

1   School of Agricultural Engineering, Jiangsu University, Zhenjiang 212013, China;
    huhuijie@stmail.ujs.edu.cn (H.H.); deerhu@ujs.edu.cn (Y.H.)
2   Jiangsu LINHAI Group, Taizhou 225310, China
3   Center for Agricultural Water Research in China, China Agricultural University, Beijing 100083, China;
    dingrsh@cau.edu.cn
*   Correspondence: yzlu@ujs.edu.cn

**Abstract:** Seasonal drought happens frequently in the lower slope hilly areas of China, which leads to a huge economic loss to China's famous tea production. An accurate determination of the evapotranspiration (*ET*) value of different seasons can provide a crucial decision parameter for irrigation management. The surface renewal (*SR*) method is an accurate and inexpensive method compared with the eddy covariance (*EC*) method, which is widely used to calculate the sensible heat flux (*H*). The latent heat flux (*LE*) evapotranspiration can be estimated indirectly when combined with the energy–balance equation. This research investigated the traditional and improved calculation methods of *H* ($SR_{snyder}$ and $SR_{chen}$), based on the surface renewal theory, over a tea field for one year. The calculation accuracy was obtained from the statistical analysis between the *SR* and *EC* methods. Different months' applicability was evaluated to determine the best calculation method for the tea field. The traditional calculation method ($SR_{snyder}$) is based on the van slope model using the second, third, and fifth structure function. The improved *SR* model ($SR_{chen}$) introduces a third order temperature function and friction velocity for calculation. The results indicate that $SR_{chen}$ shows a good calculation accuracy of *H* in the spring seasons (February to April), summer (May to July), and autumn (August to October). The determination coefficients of regression analysis ($R^2$) ranges were [0.66, 0.88] with most values greater than 0.8. The root mean square error (*RMSE*) ranges were W/m². However, during this period, $SR_{snyder}$ had a poor calculation accuracy of *H*, and the range of $R^2$ was [0.45, 0.74] with the *RMSE* ranges of [32.28, 63.25] W/m². In the winter (November to January), the calculation accuracy of both models was relatively low with $R^2$ almost 30% lower than that of other seasons. Therefore, this study suggests the use of the $SR_{chen}$ method to estimate the *H* of a tea field ecosystem in the low slope hilly area of the Yangtze River region in the spring, summer, and autumn. While in the winter, the $SR_{snyder}$ method is recommended.

**Keywords:** surface energy flows; surface renewal; eddy covariance; evapotranspiration

## 1. Introduction

Tea (*Camellia sinensis*) originates from China and is normally grown in the hilly regions. A lower hilly region located in the Yangtze River is the main production of famous teas with a higher economic value than other regions. Climate changes have caused seasonal drought in the recent years and has caused huge economic losses to the famous teas [1]. Knowledge of the energy exchanges among tea, soil, and atmosphere can adequately estimate the water demand of tea trees under the impact of water stress on crop performance [2]. An applicable energy flux measurement method for agricultural ecosystems vary with different time scales and climatic conditions.

The eddy covariance (*EC*) method performs well in most agricultural and forestry ecosystems for the estimation of energy flux. It is based on the eddy covariance theory

and calculates time-averaged covariances by directly measuring the fluctuations of the atmospheric boundary layer of water vapor, temperature, and wind speed to obtain energy flux [2]. In 1930, the eddy covariance technique was first applied to calculate the vertical flux of horizontal momentum by recording signals that were proportional to the vertical and horizontal wind speed components [3]. In the early 1980s, the development of ultrasonic anemometers and fast-response humidity sensors greatly promoted the measurement of water and heat fluxes using the eddy covariance method. In the 1990s, high-precision $CO_2$-$H_2O$ infrared analyzers were conducted to synchronously monitor the water vapor and $CO_2$ fluxes, which was a significant breakthrough and innovation in the *EC* observation technology [4,5]. The *EC* method has a high accuracy for estimating the sensible heat and latent heat flux, but the required instruments and sensors are expensive [6,7].

The surface renewal (*SR*) method, considered to be a feasible complement to the *EC* method, is also based on the eddy covariance (*EC*) theory [8,9]. The *SR* method was originated to study heat transfer between gases and liquids [5,10]. With the introduction of the concept of variable time intervals and turbulent statistics [11], it was applied to analyze the transient flow and momentum transfer [12,13]. Currently, the *SR* method is considered to be the best alternative to the *EC* [14] for estimating the sensible heat flux (*H*) with a low cost. When combined with the energy–balance equation, the latent heat flux (*LE*) evapotranspiration can be estimated indirectly, which is useful for irrigation guidance [15].

Suvočarev, K. et al. (2013) utilized the *SR* method to estimate the *H* and *LE* over varied crop surfaces and concluded that the *SR* method was a dependable alternative to the *EC* for estimating turbulent fluxes associated with irrigated agriculture [16]. Shapland, T.M. et al. (2014) compared the time-domain and frequency-domain methods for compensating the frequency response of a thermocouple, and proposed a new method for compensation in the lag domain. The study concluded that the $\alpha$ calibrations reported in the literature were primarily influenced by the frequency response characteristics of the thermocouple [9]. Similarly, Mekhmandarov, Y. et al. (2015) conducted reference measurements of *H* using the *SR* technique and found that the sampling frequency could be reduced to as low as 2 *Hz* without a significant impact on performance in shading screenhouses at a low cost [17]. Suvočarev, K. et al. (2019) applied the high-frequency (20 Hz) scalar data, turbulence, and similarity parameters in the *SR* method to calculate turbulent *H*, *LE*, and $CO_2$ ($F_c$) flux. They found that the *SR* method could be an alternative to the *EC* method as it eliminated the need for sonic anemometry [18].

When an air mass passes over a surface, energy is transferred between the two and the air is heated or cooled. The air mass then leaves the surface and a new one takes its place [19]. Under that kind of repetitive natural process, the observed wave pattern of temperature appears consistent when a high-frequency temperature measurement is conducted at a point above the surface. When the cold air reaches the surface, the high-frequency temperature of the surface appears to sharply drop. The temperature will be still for a period before the air mass is heated, and then gradually rises [20,21]. Within a few seconds, the warm air mass will be upward and a new cold air mass will replace it, completing the energy exchange process [22].

Chen et al. (1997) [23,24] redefined Snyder's model ($SR_{snyder}$) using the dimensional analysis and plant canopy turbulence theory. Their model directly estimates *H* from the third-order temperature function and friction velocity. The main difference between Chen's model ($SR_{chen}$) and Snyder's model is that Chen's model considers the temperature change that occurs over a period of time when a new air parcel replaces the old one, rather than an instantaneous process, although the duration of this period is very short. The friction velocity used in their model not only corrects this error, but also corrects the time delay caused by using thermocouples of different thicknesses [9]. Thinner thermocouples have less time delay, but are also more prone to breakage. Using the friction velocity can correct this problem, allowing for the use of slightly thicker thermocouples without affecting the final measurement results [25].

The objective of this research is to evaluate the accuracy of different *SR* methods in calculating *H* over different time periods at the interannual scale. Two different *SR* models (*SR_{snyder}* and *SR_{chen}*) based on the surface renewal theory were applied in this research within a tea field for a one year experiment. The calculation accuracy was obtained from the statistical analysis between the *SR* and *EC* methods. Different months' applicability was evaluated to determine the best calculation method for the tea field.

## 2. Materials and Methods

### 2.1. Experimental Site

The experimental tea field was located in the middle–lower Yangtze River region, east of China where was the main production areas of famous tea. The topography of the experimental site is a hilly ground with an average altitude of 18.5 m (latitude 32°01′35″ north (N), longitude 119°40′21″ east (E)). It has a moderate subtropical climate with a mean annual precipitation of 1029.1 mm and an average annual temperature of 15.5 °C. The annual reference evapotranspiration ($ET_0$) was 892.24 mm, which was observed over the last 55 years (1961–2015). The precipitation is concentrated between April and October, accounting for 80% of the annual rainfall. The average relative humidity is 76%, the annual accumulated sunshine hours total 2051.7 h, the frost-free period is 239 days, and the annual average wind speed is 3.4 m/s. The soil in the tea plantation is a yellow-brown loam with a density of 1.7 g/cm$^3$ [26] (Figure 1).

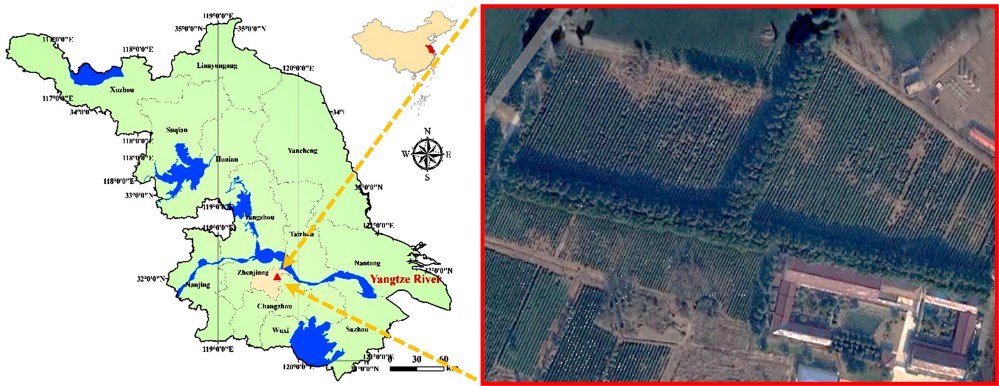

**Figure 1.** Location of the experimental tea plantation.

Tea has a significantly different growth rhythm from conventional field crops such as wheat and corn [27]. As a perennial evergreen woody plant, it grows year-round except during the winter. In this research, the sampled tea cultivar was Maolv, which was eight years old and was harvested in March and April. To avoid severe water stress during the summer, canopy pruning is carried out in May. The highest temperature of the year, reaching 43.3 °C, occurred at the end of July and the beginning of August, which caused a serious drought disaster.

### 2.2. Micrometeorological Data Collection

The surface renewal (*SR*) observation system consisted of a CR3000 data logger (CR3000, Campbell Scientific, Logan, UT, USA), an air temperature and humidity sensor (HC2S3, Campbell Scientific, Logan, UT, USA), a four-component net radiation sensor (CNR4, Kipp &Zonen, Delftechpark, Delft, The Netherlands), a 3-D sonic anemometer (CSAT3, Campbell Scientific, Logan, UT, USA), two thermocouples (Type E, OMEGA, Norwalk, CT, USA), a soil heat flux plate (HFP01SC, Hukseflux, Delftechpark, Delft, The Netherlands), and a soil temperature and moisture sensor (Hydra Probe II, Stevens, Portland, OR, USA).

The *EC* system consisted of an open-circuit $CO_2$/$H_2O$ gas analyzer (EC150, Campbell Scientific, Logan, UT, USA) and a 3-D sonic anemometer (CAST3, Campbell Scientific,

Logan, UT, USA). EC150 is an open-path analyzer specifically designed for eddy covariance carbon and water flux measurements. The *EC* system measured at a height of 2.5 m above the soil surface, and the original data was stored at a sampling frequency of 10.0 Hz. To analyze the flux transport characteristics in the tea field more accurately, the half-hourly *H* and *LE* data obtained from the *EC* system were processed and corrected by using the Eddypro 6.1. The corrections included subsurface, density effects, and meteorological factors (Figure 2).

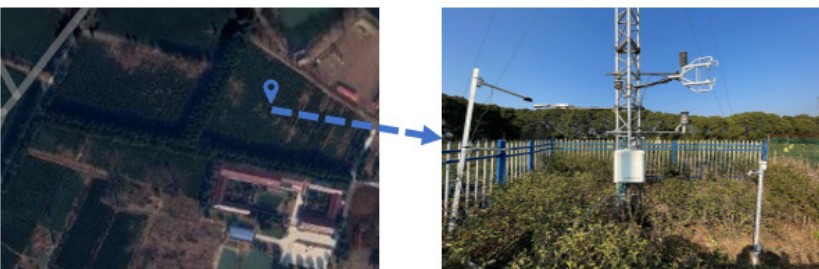

**Figure 2.** *EC* and *SR* observation systems.

Two thermocouples were placed at a height of 1.2 m (tea canopy height). The net radiometer sensor was positioned 2.5 m above the canopy to measure the downward and upward shortwave radiations ($R_{sd}$ and $R_{su}$), as well as the downward and upward longwave radiations ($R_{ls}$ and $R_{lu}$). A 3-D sonic anemometer was also set at a height of 2.5 m to measure wind speed ($u$) and direction. The soil heat flux sensor was placed 5.0 cm below the canopy to measure the surface soil heat flux ($G$). Additionally, soil temperature and moisture sensors were positioned at two different depths (2.0 cm and 8.0 cm) beneath the soil heat flux plate to measure surface soil temperature ($T_{soil}$) and soil volumetric water content ($VWC$). The sensors were set to acquire data at a frequency of 10 Hz. The data set includes turbulence data obtained through both the eddy covariance method and the surface renewal method. Other required data, such as net radiation and soil heat fluxes, were collected from 1 January 2022 to 31 December 2022. However, due to system maintenance and occasional power outages, some data were missing during certain periods, which were excluded from the final calculation.

### 2.3. Traditional Surface Renewal Method on Calculating the H

In stable and unstable atmospheric conditions, the temperature slope is characterized by the amplitude ($a$) and inverse slope frequency ($l + s$), as shown in Figure 3. If the air above is colder than the air at the surface (unstable), high-frequency temperature data will sharply decrease when an air mass sweeps over the surface. This is followed by a period of calm ($s$) and then a period of instability ($l$) as the air mass is heated by the surface and the temperature gradually increases. Typically, within a few seconds, the warm air mass jets upward and is replaced by another cool air mass from above [28].

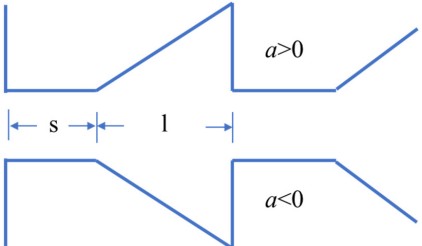

**Figure 3.** Schematic temperature ramps with amplitude ($a > 0$) indicating unstable and ($a < 0$) stable atmospheric conditions. The frequency of the inverse lapse rate ($l + s$) in seconds is the sum of the quiescent period ($s$) and the ramp ($l$).

Paw, U et al. (1995) expressed sensible heat flux density ($H$) in terms of the change in heat content with time ($dT/dt$) as in [29]:

$$H = \alpha \rho C_p \frac{dT}{dt} \frac{V}{A} \tag{1}$$

where, $\alpha$ is the proportionality factor; $\rho$ is the air density, kg·m$^{-3}$; $C_p$ is the specific heat of air, J·kg$^{-1}$ K$^{-1}$; $V/A$ is the volume of air per unit area under the canopy height. They assumed the parcel height equaled the canopy height ($z_c$), therefore [28]:

$$H = \alpha \rho C_p \frac{a}{(l+s)} z_c \tag{2}$$

where, $a$, $l$, and $s$ are the defined phases and they can be calculated by the equation proposed by van de Griend, A.A. and Owe, M. [30], a model that allows the use of second-order, third-order, and fifth-order temperature equations to calculate the values of $a$ and $(l+s)$. Because the true value of the $H$ is calculated by the *EC* technique, the optimal time interval should be 30 min [5], both for calculating the correction factor, $\alpha$, or for comparing accuracy. The structure function, $S^n(r)$, is defined as:

$$S^n(r) = \frac{1}{m-j} \sum_{i=1+j}^{m} (T_i - T_{i-j})^n \tag{3}$$

where, $m$ is the number of data points collected during the 30 min measurement interval, $j$ is the sample lag, and $T_i$ is the $i$th temperature sample, °C. The estimation of the amplitude, $a$, during a particular 30-min interval is obtained by solving the following Equation (4) for the real numbers:

$$a^3 + pa + q = 0 \tag{4}$$

where, the value of $p$ and $q$ in Equation (4) are calculated as:

$$p = 10S^2(r) - \frac{S^5(r)}{S^3(r)} \tag{5}$$

$$q = 10S^3(r) \tag{6}$$

After finding the value of $a$, then the value of $(l+s)$ is equal to:

$$(l+s) = -\frac{a^3 r}{S^3(r)} \tag{7}$$

The value of $H$ can be obtained by combining Equations (3)–(7).

### 2.4. Improved SR Method on Calculating the H

The improved surface renewal method is based on Equation (2) combined with the theory related to digital signal processing, taking fully into account the property that the temperature change occurs in a very short time rather than instantaneously [16]. This model is estimated by using a third-order temperature function and friction velocity. The specific calculation process is done using the dimensional analysis and completed through the following equations:

$$H = \begin{cases} -\alpha \beta^{\frac{2}{3}} \gamma \rho C_p \left[ \frac{S^3(\Delta t_m)}{\Delta t_m} \right]^{\frac{1}{3}} u^{*\frac{2}{3}} \frac{z}{h^{\frac{2}{3}}} & for \quad 0.2h < z \leq h + 2(h-d) \\ -\alpha \beta^{\frac{2}{3}} \gamma \rho C_p \left[ \frac{S^3(\Delta t_m)}{\Delta t_m} \right]^{\frac{1}{3}} u^{*\frac{2}{3}} \frac{z}{(z-d)^{\frac{2}{3}}} & for \quad z > h + 2(h-d) \, or \, z \leq 0.2h \end{cases} \tag{8}$$

where, $\alpha$, $\beta$, and $\gamma$ are the empirical coefficients; $\Delta t$ is determined for each 30-min interval as the $\Delta t$ for which the absolute value of $S^3(\Delta t)/\Delta t$ is maximum (denoted by $\Delta t_m$); $u^*$ is friction velocity, m·s$^{-1}$; $h$ is the canopy height, m; and $d$ is the zero-plane displacement height, m. In this case, the following expressions were defined using the dimensional analysis and plant canopy turbulence theory:

$$\frac{a}{\tau^{\frac{1}{3}}} = -\gamma \left[ \frac{S^3(\Delta t_m)}{\Delta t_m} \right]^{\frac{1}{3}} \tag{9}$$

The $\gamma$ in the above Equation (9) is an empirical coefficient, and:

$$\frac{1}{\tau} = \begin{cases} \beta \frac{u^*}{h} & for\ 0.2h < z \leq h + 2(h-d) \\ \beta \frac{u^*}{(z-d)} & for\ z > h + 2(h-d)\ or\ z \leq 0.2h \end{cases} \tag{10}$$

where $\beta$ is the empirical coefficient and its calculation is obtained by taking the slope of linear regression between $\frac{1}{\tau}$ and $\frac{u^*}{h}$. The first expression is applicable to the canopy and roughness sublayers and the second expression is applicable to the inertia sublayers. The focus of this paper is on the tea canopy, thus Equation (2) is chosen.

Compared with the $SR_{snyder}$ method, only two empirical coefficients, $\beta$ and $\gamma$, are suitable for this crop and should be calculated in advance for the $SR_{chen}$ method. At the end of each ramp, a sudden change in temperature occurs, which is considered instantaneous by the traditional approach, but is considered to be completed in a very short period of time in the computational. When the sampling frequency of the thermocouples was set to 80 Hz, there were 3–5 temperature sampling points in a short time with the energy transfer process, which was indeed a large but not infinite slope [31].

### 2.5. Eddy Covariance (EC) Method on Calculating the H

The original data is processed using flux data with a 30-min interval. There is a significant amount of transient noise, also known as outliers, in the turbulent data, which can significantly affect the covariance values [32]. These outliers are mainly caused by the interference of complex environmental factors on the ultrasonic sensor readings [21]. The system removes outliers using Virkler's method. The coordinates are subjected to quadratic rotation (tilt correction) using the plane-fitting method proposed by Dr. Kyaw Tha Paw U of the University of California, Davis [5]. Frequency loss is corrected by spectral correction, transfer function for temporal-averaging correction, measurement path-/volume-averaging correction, time-constant correction, and sensor-spacing separation correction [26]. Sensible heat flux is related to the virtual temperature, $Ts$, and the improved sonic temperature gradient method is used to revise the virtual temperature of the sensible heat flux. The approximate equation is given as:

$$H = \rho C_p \overline{w'T'} \tag{11}$$

where, $w'$ is vertical wind speed fluctuation, m/s; $\rho$ is water vapor density, kg/m$^3$; and $T'$ is air temperature fluctuation, °C.

$$\rho = \frac{P}{[R_d \times (T_a + 273.15)]} + \rho_v \tag{12}$$

$$C_p = C_{pd}(1 + 0.84q) \tag{13}$$

where, $P$ is atmospheric pressure, Pa; $R_d$ is dry air gas constant, 287 J·kg·K$^{-1}$; $T_a$ is air temperature, °C; $C_{pd}$ is the specific heat of dry air at constant pressure, 1005 J·kg$^{-1}$ K$^{-1}$; and $q$ is specific humidity, kg·kg$^{-1}$. Although the $EC$ system directly measures the density of different particles, such as vertical wind speed, air temperature, and high-frequency pulsation of water vapor density, and the instantaneous fluctuations of each component are

large, the instantaneous values fluctuate around a specific value with relative stability in the time-averaged range (half an hour is taken for this experiment).

*2.6. Determination of the $\alpha$, $\beta$, $\gamma$, $u^*$ and Length Scale h, $z - d$ for Surface Renewal Method*

The temperature fluctuations at the tea canopy were recorded by a type E thermocouple and used to calculate the $H$. Water density and specific heat of air, both essential for calculating the energy flux, were determined using a 3-D sonic anemometer. Ten days were randomly selected from each of the months of January, April, July, and October to calculate the empirical coefficients of different models. All data were used to evaluate the reliability of the two methods.

Both the $SR_{snyder}$ and the $SR_{chen}$ have empirical coefficients. It was the slope which was calculated by the linear regression of $H_{EC}$ and $H_{SR}$ with a 30-min interval [5,9,33,34].

In the calculation of the $SR$ method considering the friction velocity, two more empirical coefficients, $\beta$ and $\gamma$, were added compared to the traditional method. The temperature ramp model is calculated according to the method considering the friction velocity by obtaining the values of $\frac{S^3(\Delta t)}{\Delta t}$ to obtain the values of $\alpha$ and $\tau$. Then, the value of $\beta$ is the slope of the linear regression between $\frac{1}{\tau}$ and $\frac{u^*}{h}$, and the value of $\gamma$ is the average of the ratio of $\frac{a}{\tau^{\frac{1}{3}}}$ and $-\left(\frac{S^3(\Delta t_m)}{\Delta t_m}\right)^{\frac{1}{3}}$ [24].

In the Monin–Obukhov similarity theory, the Monin–Obukhov length ($L$) and friction velocity ($u^*$) are two vital characteristic scales to measure the near-surface turbulence characteristics. It can be calculated by [35] as:

$$L = \frac{-\rho c_p T_a u^{*3}}{kg\overline{w'T'}} \tag{14}$$

$$u^* = \left(\overline{u'w'}^2 + \overline{v'w'}^2\right)^{\frac{1}{4}} \tag{15}$$

where, $L$ is the Monin–Obukhov length, m; $u^*$ is the friction velocity, m/s; $k$ is the Karman constant, taken as 0.4; and $g$ is the acceleration of gravity, taken as 9.8 m/s$^2$. The friction velocity is calculated from the above Equation (15) using the wind speed data measured by CSAT3, after coordinate rotation.

The inertial sublayer is expressed as the displacement height ($z - d$), with $d$ being 0.67 times the height of the tea plant.

*2.7. Model Evaluation Based on Statistical Analysis*

Energy closure is an important indicator for assessing the quality of flux calculation models and is a crucial aspect of studying water, heat transfer, and energy balance in ecosystems. Energy closure is defined as the percentage of available energy (the sum of latent and sensible heat) over usable energy (the difference between net radiation and soil heat flux) during a certain period, also known as the energy balance ratio (*EBR*). In the case of absolute energy closure in a system, the energy closure is 1 in an ideal state. In this study, *EC* was used to estimate $H$ and $LE$ for the 12 months of 2022 and to analyze the energy closure to evaluate the model. Based on 30-min flux data, the ratio of effective energy to usable energy in the system was calculated, and the energy closure was statistically averaged. *EBR* can be expressed as:

$$EBR = \frac{H + LE}{R_n - G} \tag{16}$$

The estimated $H$ can be evaluated using the two *SR* methods based on the root mean square error (*RMSE*), mean absolute error (*MAE*), systematic root mean square error (*SRMSE*), and determination coefficient ($R^2$).

## 3. Results

### 3.1. Analysis of Energy Closure for EC Method

As shown in Figure 4, the monthly average of *EBR* in the year of 2022 dynamically changed with the seasons. In the spring, summer, autumn, and winter, the seasonal average of *EBR* was 0.85, 0.67, 0.57, and 0.53, respectively. It can be seen that the *EBR* is highest in spring, followed by the summer, then autumn and winter. Therefore, different seasons should be corrected using their corresponding average energy closure.

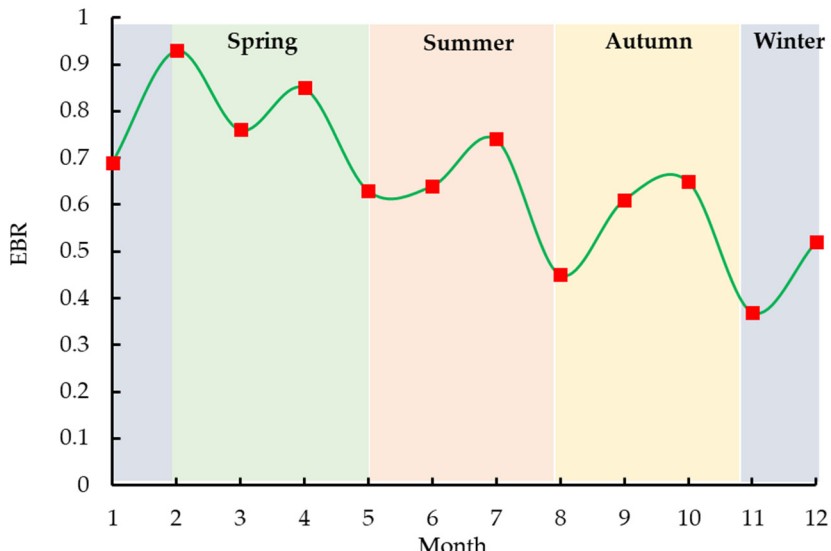

**Figure 4.** Energy balance ratio of different months for a tea field ecosystem.

### 3.2. Calibration of the SR Model

For the $SR_{snyder}$ method, the $\alpha$ coefficient needs to be calculated for each season. In this research, the $\alpha$ coefficient for the spring, summer, autumn, and winter was 1.52, 0.82, 1.12, and 1.31, respectively. For the $SR_{chen}$ method, three empirical coefficients, which were $\alpha$, $\beta$, and $\gamma$, needed to be calculated. After calculation, the value of $\beta$ was 0.296. The value of $\gamma$ was calculated to be 0.999; therefore, it was taken as 1 for subsequent calculations. The value of $\alpha$ does not differ significantly among the different months, ranging from 0.82 to 0.93, thus an average value of 0.879 for the entire year was selected.

### 3.3. Validation of the SR Model

The statistical analyses between the *H* calculated using different models ($H_s$ and $H_c$ represent the H calculated from the $SR_{snyder}$ and $SR_{chen}$ methods) and $H_{EC}$ (represent the H calculated from the *EC* method) for each month of the full year are shown in Table 1. For the statistical analysis between *Hs* and $H_{EC}$, the $R^2$ ranges were [0.34, 0.76] and *RMSE* ranges were [30.79, 69.50] W/m$^2$. The *MAE* and *SRMSE* ranges were [16.92, 38.95] W/m$^2$ and [30.15, 69.19] W/m$^2$, respectively. For *Hc* and $H_{EC}$, the $R^2$, *RMSE*, *MAE*, and *SRMSE* ranges were [0.52, 0.88] W/m$^2$, [30.05, 69.22] W/m$^2$, [15.89, 43.69] W/m$^2$, and [29.06, 58.18] W/m$^2$, respectively.

It can be seen that the $SR_{chen}$ method showed a good calculation accuracy of *H* in the seasons of spring (February to April), summer (May to July), and autumn (August to October). The $R^2$ ranges were [0.66, 0.88] with most values greater than 0.8. The *RMSE* ranges were [34.15, 69.22] W/m$^2$. The $SR_{chen}$ method had a better performance in both coherence and slope in the seasons of spring, summer, and autumn. However, during this period, $SR_{snyder}$ had poor calculation accuracy for *H*, and the range of $R^2$ was [0.45, 0.74] with the *RMSE* ranges within [32.28, 63.25] W/m$^2$.

**Table 1.** Statistical analysis of $H$ calculated by $SR_{snyder}$ and $SR_{chen}$ for different months.

| | Month | RMSE (W·m$^{-2}$) | MAE (W·m$^{-2}$) | SRMSE (W·m$^{-2}$) | $R^2$ |
|---|---|---|---|---|---|
| $H_s$ vs. $H_{EC}$ | January | 30.79 | 18.26 | 30.15 | 0.76 |
| | February | 32.28 | 16.92 | 30.90 | 0.67 |
| | March | 49.93 | 26.96 | 47.90 | 0.58 |
| | April | 48.10 | 29.73 | 47.48 | 0.56 |
| | May | 63.35 | 32.26 | 62.27 | 0.45 |
| | June | 63.25 | 38.95 | 56.45 | 0.71 |
| | July | 47.8 | 38.31 | 45.73 | 0.46 |
| | August | 36.13 | 22.60 | 35.93 | 0.56 |
| | September | 41.21 | 23.58 | 39.02 | 0.74 |
| | October | 41.50 | 24.94 | 40.41 | 0.69 |
| | November | 69.50 | 27.56 | 69.39 | 0.34 |
| | December | 53.64 | 35.38 | 51.53 | 0.46 |
| $H_c$ vs. $H_{EC}$ | January | 30.05 | 18.42 | 29.06 | 0.64 |
| | February | 30.75 | 15.89 | 29.54 | 0.88 |
| | March | 35.53 | 21.96 | 33.81 | 0.88 |
| | April | 38.14 | 22.85 | 36.01 | 0.83 |
| | May | 55.59 | 33.61 | 51.44 | 0.66 |
| | June | 69.22 | 43.69 | 58.18 | 0.88 |
| | July | 46.66 | 28.96 | 40.69 | 0.81 |
| | August | 34.15 | 20.99 | 31.63 | 0.66 |
| | September | 48.39 | 27.51 | 43.84 | 0.85 |
| | October | 41.39 | 24.54 | 39.14 | 0.86 |
| | November | 38.16 | 21.22 | 36.31 | 0.44 |
| | December | 44.60 | 29.91 | 41.18 | 0.52 |

In the winter (November to January), the air temperature around the tea canopy was extremely low and radiation frost events occurred frequently in the months of November and December. Although the correlation was better and the error was relatively small, the value of the slope was significantly worse than that of $SR_{snyder}$ method, especially in January when the correlation of the $SR_{snyder}$ method was better than the other one.

## 4. Discussion

We have preliminarily investigated the performance of using the $SR_{snyder}$ method to estimate the $H$ over a tea field in the winter and summer, and found that the result of summer was better than that of winter [36]. Snyder et al. (1996) measured high frequency temperature data at five heights over 0.1 m tall for fescue grass, which showed a good estimation on $H$ if data were not collected too close to the canopy top or above the fully adjusted boundary layer [28]. Both Snyder (1996) and our previous research did not select a complete year for the experiments, which cannot demonstrate the performance of $SR_{snyder}$ under annual conditions. For the $SR_{chen}$ method, Chen et al. (1997) only validated in the summer for three different scenarios which were forest, bare soil, and straw, demonstrating that it performs better than the $SR_{snyder}$ method [22]. However, as tea plant belongs to the shrub family, the $SR_{chen}$ method had never been applied in a tea field before. This research

was the first trial on the annual evaluation of two SR methods on calculating the *H* over a tea field ecosystem [37].

As shown in Figure 5, the *SR* method had some values close to zero in November. On the one hand, this was due to the cold wave causing a significant drop in temperature, which led to a decrease in the stability of the thermocouple. The *SR* method requires thermocouple data, resulting in abnormal values. At the same time, the *EC* method does not use thermocouple data, resulting in a low correlation slope between the two methods. On the other hand, November had more rainfall in the tea field, which made the estimation of *H* by the *SR* method unreliable. Although the data on rainy days has been excluded, high humidity after rainfall can still affect the results.

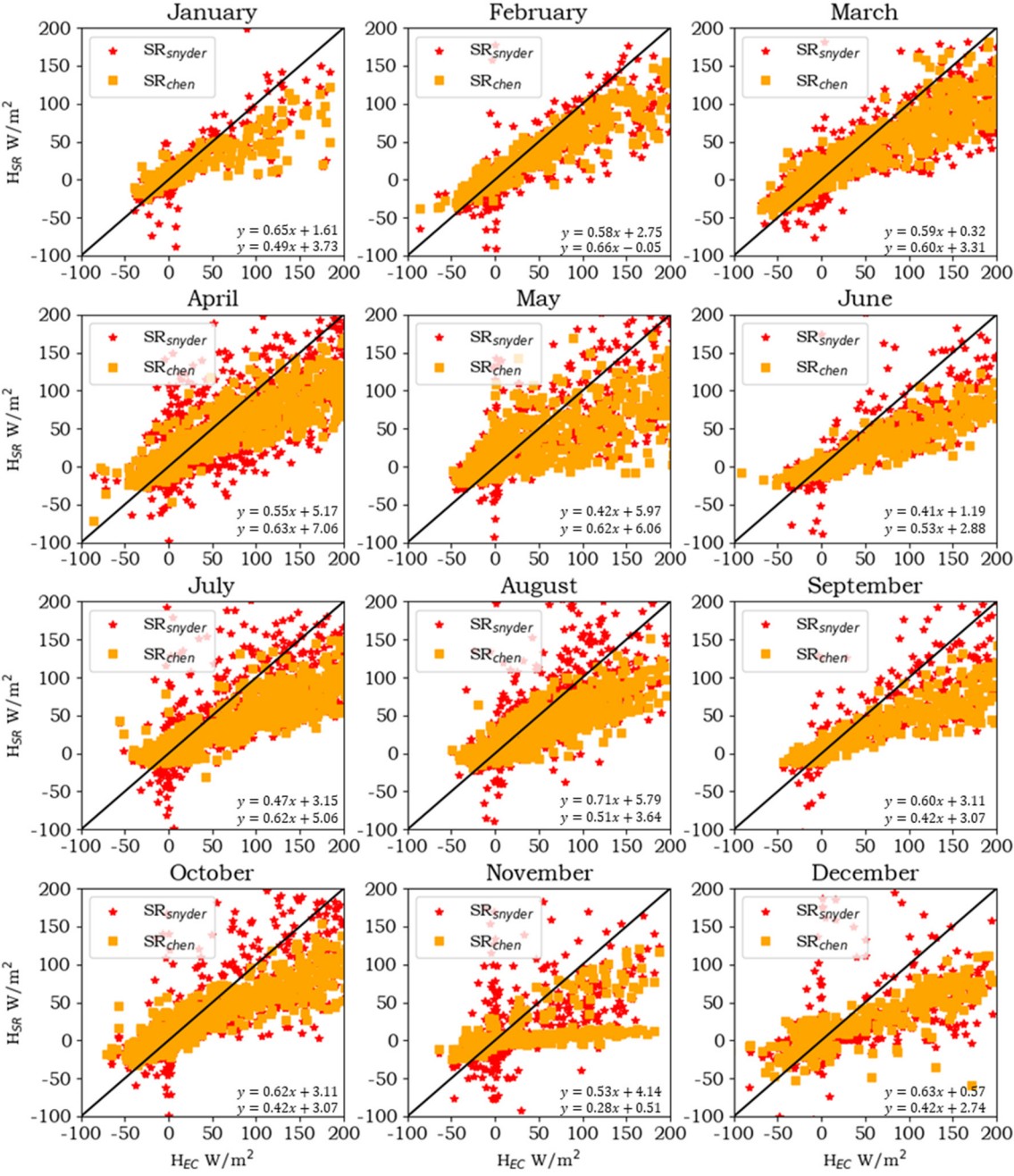

**Figure 5.** The correlation between sensible heat fluxes calculated using two different surface renewal methods, and sensible heat flux calculated using the eddy covariance method. Also shown are the results of the linear regression. All values of *p* are lower than 0.05.

A cubic equation needs to be solved for each $H$ value when using the $SR_{snyder}$ method to estimate the $H$. If the data is unstable, the value of the third-order temperature structure function may be close to zero. Since it is used as a denominator in the calculation of $p$ in Equation (5), this will cause distorted points in the calculation. However, the $SR_{chen}$ method does not require a cubic equation for each time calculation. Instead, the third-order temperature structure function is needed to multiply with other terms. Chen also pointed out that in unstable atmospheric conditions, $H$ was calculated from the third-order temperature structure function, better described as the half-hour sensible heat flux changes in the canopy and rough sublayer than the $SR_{snyder}$ method. Wilson also noted that the $SR_{snyder}$ method exhibited overestimation of negative $H$ and underestimation of positive $H$ observed at the ground level [38]. This may be related to the amplification of errors in $a$ in the van model within $(l + s)$ [24], while $SR_{chen}$ avoids the uncertainty in $a$ and $(l + s)$.

The friction velocity of the canopy was in fact obtained from wind profile calculations under normal temperature conditions, which led to significant calculation errors in the cold winter. Holwerda proposed that the roughness above the canopy is about 17% higher than that of the canopy level, leading to increased errors in $H$ estimation [35]. Additionally, the reliability of the thermocouples may reduce during the winter due to the interference of low temperature. An uncertainty power supply from the solar panel may also cause the fluctuations in voltage and further in the winter which reduced the reliability of the data [39]. These factors are inevitable objective factors that contribute to the decreased reliability of instruments during the winter.

## 5. Conclusions

This research investigated the traditional and improved calculation methods of $H$ ($SR_{snyder}$ and $SR_{chen}$) based on the surface renewal theory within a tea field for a one-year experiment. The calculation accuracy was obtained from the statistical analyses between the $SR$ and $EC$ methods. Different months' applicability was evaluated to determine the best calculation method for the tea ecosystem.

This study has certain limitations. Firstly, the $SR$ method itself has distortion points generated by calculation and the not low occurrence rate causes many points to be removed, resulting in much fewer available points for use, which has a certain impact on the correlation with the $EC$ method. Secondly, this study only collected one-year data. Actually, there are differences in climate between different years in the same season and it is necessary to observe for another two years to obtain more reliable results. Thirdly, due to the limitations of the experimental field conditions, it does not meet the requirement of being open within a certain range which has an impact on eddies. Finally, this study only set up an $EC$ meteorological station in a flat place over a hilly tea field. In the future research, we will set up meteorological stations under different slope conditions to study the possible impact of slope on the models.

In this research, the $SR_{chen}$ method showed a good calculation accuracy of $H$ in the seasons of spring (February to April), summer (May to July), and autumn (August to October). The $R^2$ ranges were [0.66, 0.88] with most values greater than 0.8. Additionally, the $RMSE$ ranges were [34.15, 69.22] $W/m^2$. However, during this period, $SR_{snyder}$ had a poor calculation accuracy of $H$ and the range of $R^2$ was [0.45, 0.74] with the $RMSE$ ranges of [32.28, 63.25] $W/m^2$. In the winter (November to January), the calculation accuracy of both models was relatively low with $R^2$ almost 30% lower than that of other seasons.

Therefore, the traditional surface renewal model ($SR_{snyder}$) is recommended to estimate the sensible heat flux during the winter, and for the rest of the months of the year, the improved model ($SR_{chen}$) is recommended.

**Author Contributions:** Conceptualization, H.H., Y.L. and Y.H.; methodology, H.H. and Y.L.; software and data collection, H.H. and Y.L.; validation, Y.L.; investigation, H.H. and Y.L.; writing—original draft preparation, H.H. and Y.L.; writing—review and editing, Y.H. and R.D.; supervision, Y.L. and Y.H.; project administration, Y.L. and Y.H.; funding acquisition, Y.H. and Y.L. All authors have read and agreed to the published version of the manuscript.

**Funding:** This research was funded by the Key Research and Development Program of Jiangsu Province (BE2021340), the China and Jiangsu Postdoctoral Science Foundation (2022M711396 and 2021K614C), the project of Key Laboratory of Modern Agricultural Equipment and Technology, Jiangsu University (MAET202119), the project of Jiangsu Province and Education Ministry Co-sponsored Synergistic Innovation Center of Modern Agricultural Equipment (XTCX2013), the Natural Science Foundation of the Jiangsu Higher Education Institutions of China (21KJB210019), and the Priority Academic Program Development of Jiangsu Higher Education Institutions (PAPD-2018-87).

**Data Availability Statement:** Not applicable.

**Acknowledgments:** The principal author is extremely indebted to all sensor providers and their staff for establishing the micrometeorological observation stations used in the current study. We also wish to express our gratitude to the School of Agricultural Engineering, Jiangsu University for providing essential infrastructure and instruments without which this work would not have been possible. We would like to thank the anonymous reviewers for their precious attention.

**Conflicts of Interest:** The authors declare no conflict of interest.

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
