# Peer review of "Evaluation of Two Surface Renewal Methods for Calculating the Sensible Heat Flux over a Tea Field Ecosystem in Hilly Terrain"

_agronomy, doi:10.3390/agronomy13051302_

Round 1

Reviewer 1 Report

Dear Authors,

The manuscript needs some adaptations, before final acceptance. All corrections, suggestions and recommendations are attached in the attached file.

Reviewer 2 Report

The authors present an interesting study where they explore an alternative to eddy covariance-based methods to estimate sensible heat flux. The study merits publication provided the manuscript improves in a number of aspects. I present some remarks hereafter; although my list is not exhaustive.

Line 38: Please check weather the two words composing the scientific name of a plant are written in capital letters. I think only the genus should be written in capital letters: Camellia sinensis

Line 67: Check the right spelling of the author you are citing.

Some references cited in the text do not appear in the list of references. For instance: Line 76 (BRK et al). There is a combined system of citing references: by number and by author family name (year). Please adopt only one.

The Surface Renewal method is the core of the article. However, no verbose description of the principles of this method is presented and its differences with eddy covariance are presented. This is an important missing point in the article.

Figure 2: Pictures of both systems (EC and SR) are presented. It would be useful to see how they are located with respect to each other and with respect to other landscape features. This could help understand/interprete the results. Consider preparing a map showing the location of both measurement stations and other components of the landscape.

Line 165: “….proposed by van [29]….”. van?

Line 206: typo: covariacne

Check the hyphenation rules you are following.

Figure 4: What happened in november in the SRchen method? There seems to be an artifact producing values around 0.

The reference to evaluate the performance of the alternative method is the eddy covariance method. It can be useful to discuss the uncertainties the EC method may have. That can only help in the interpretation of the results.

Reviewer 3 Report

Title: last words “low slope hill” need to rephrased

Citations in some places do not follow the journal format

Define abbreviations before first time use

Use consistent terms throughout the paper. For example, soil moisture or soil humidity

Equations 1-6 are not well defined and/or correlated.

Not clear how Equitation 8 can lead to determining gamma?

The SRchen model not discussed well in the methodology compared to SRsnyder’s?

The major drawback of the paper is that the description of the methods and how they were used to fulfill the targeted objectives need to be more focused.
